

# Potential role of photobiomodulation as a prevention and treatment strategy for radiation induced fibrosis: a review of effectiveness and mechanisms

Rachita Gururaj[1], Betty Thomas[1], Manur Gururajachar Janaki[2],
Vinay Martin D'sa Prabhu[3], Rakesh Nagaraju[4], Stephen Rajan Samuel[5] and
Sundar Kumar Veluswamy[1]

[1] M.S. Ramaiah College of Physiotherapy, M.S. Ramaiah University of Applied Sciences, Bengaluru, Karnataka, India
[2] Department of Radiation Oncology, M.S. Ramaiah Medical College, M.S. Ramaiah University of Applied Sciences, Bengaluru, Karnataka, India
[3] Department of Radiology, M.S. Ramaiah Medical College, M.S. Ramaiah University of Applied Sciences, Bengaluru, Karnataka, India
[4] Department of Oral Medicine and Radiology, Faculty of Dental Sciences, M. S. Ramaiah University of Applied Sciences, Bengaluru, Karnataka, India
[5] Department of Health and Life Sciences, College of Arts & Sciences, Mount Vernon Nazarene University, Ohio, United States of America

## ABSTRACT

**Background**. Radiation induced fibrosis (RIF) is a chronic progressive disabling side effect of radiation therapy in cancer survivors with limited therapeutic options. Photobiomodulation therapy (PBMT) is being propagated as a non-invasive therapeutic option but has limited evidence. This scoping review aims to summarize the effects and mechanisms of PBMT in the prevention and treatment of RIF.

**Methods**. A systematic search was conducted across five databases (PubMed, Scopus, EBSCO, ProQuest, LILACS), and three other platforms (Google Scholar, ResearchGate, Academia.edu). Retrieved studies underwent independent title, abstract, full text screening and data extraction. Quality analysis was performed for human studies to assess methodological rigor.

**Results**. The review identified three studies that specifically focused on RIF. Since induction of RIF is not common for *in vitro* and *in vivo* studies, the screening was expanded to include studies targeting fibroblast cells or fibrosis of any origin. The revised strategy led to inclusion of 26 studies (nine *in vitro*, 13 *in vivo*, and four clinical studies). Of these, 20 studies focused on the prevention of fibrosis, while six addressed its treatment. Preclinical studies demonstrated the beneficial effects of PBMT at different phases of fibrosis at cellular level. Clinical studies demonstrated functional improvements. Mechanisms include modulation of inflammatory pathways, fibroblast to myofibroblast conversion, collagen production, reduction of oxidative stress, and regulation of extracellular matrix remodeling.

**Conclusion**. PBMT demonstrates potential as a non-invasive, safe therapeutic option for RIF, supported by extensive preclinical evidence. However, high-quality clinical trials are necessary to validate its clinical efficacy.

Corresponding authors
Rachita Gururaj,
rachitagururaj@gmail.com
Sundar Kumar Veluswamy,
sundark94@gmail.com

**Implication**. PBMT offers a promising intervention for managing RIF, with potential to enhance body image, self-confidence, functional abilities, and overall quality of life for cancer survivors. This review underscores the need for further research to translate these findings into clinical practice.

## INTRODUCTION

Radiation induced fibrosis (RIF) is a long-term sequela of radiotherapy affecting over 50% of cancer survivors by one year and progressively increasing to affect over 65% survivors by eight years (*Baudelet et al., 2019*). The repeated exposure to radiation during each fraction of radiotherapy leads to trigger of repetitive inflammatory processes causing excessive extracellular matrix and collagen deposition, eventually resulting in fibrosis (*Borrelli et al., 2019*; *Ejaz, Greenberger & Rubin, 2019*; *Ramia et al., 2022*). RIF initially presents as inflammation, erythema, oedema, ulcerations, fistula, and eventually as fibrosis. This fibrosis results in varied presentations such as persistent pain, reduced range of motion, xerostomia, trismus, impaired vocal quality, dysphagia, aspiration, lymphedema, hollow organ stenosis, and osteoradionecrosis, leading to significant functional limitations and impaired quality of life during the cancer survivorship (*Purkayastha et al., 2019*; *Fijardo et al., 2024*). Current treatment strategies for RIF remain predominantly in the research phase and are limited to pharmacological approaches, such as pentoxifylline, pravastatin, and vitamin E, as well as physiotherapy interventions, including ultrasound therapy, manual therapy techniques, and exercise (*Warpenburg, 2014*; *Cho & Park, 2017*; *Nogueira et al., 2022*; *Wilson et al., 2022*; *Gururaj et al., 2024*). Due to its long-term effects on functional impairment and overall quality of life in cancer survivors, RIF is attracting significant research attention (*Fijardo et al., 2024*). With its pathogenesis rooted in response of cellular repair pathways to repetitive radiation exposure (*Vallée et al., 2017*), effective management strategies need to target the cellular mechanisms to achieve therapeutic benefits.

Photobiomodulation therapy (PBMT) is a non-invasive therapeutic modality that utilizes low-level light energy (less than 500 mW) for stimulating biological processes and cellular responses. It includes light emitting diodes (LEDs), low level laser therapy and broadband light (*Anders, Lanzafame & Arany, 2015*). PBMT is known to cause increased cellular energy production, modulation of reactive oxygen species and stimulation of anti-inflammatory pathways, leading to stimulation of healthy tissue repair and wound healing (*Aggarwal & Lio, 2023*). Due to its proven beneficial effects in modulating cellular pathways, it is increasingly being adopted in clinical practice for pain relief, wound healing, and tissue regeneration (*Deana et al., 2021*; *Oyebode, Houreld & Abrahamse, 2021*). Emerging evidence also highlights its potential applications in managing fibrosis of various organs, including the lungs, liver, and cardiac tissue (*Brochetti et al., 2017*; *Ailioaie & Litscher, 2020*; *Tomazoni, Johnson & Leal-Junior, 2021*; *Feliciano et al., 2022*).

In cancer survivors, there is a growing body of evidence on safety of PBMT both during primary cancer treatment as well as during survivorship (*De Pauli Paglioni et al., 2019*; *Bensadoun et al., 2020*). Clinical practice guidelines recommendation are available for PBMT in the management of oral mucositis, radiation dermatitis, lymphedema and xerostomia (*Harris et al., 2001*; *Elad et al., 2020*; *Behroozian et al., 2023*; *Hong et al., 2024*). There has been a growing body of literature advocating PBMT for the management of RIF, but such recommendations are limited to narrative reviews and position papers based on consensus (*Tam et al., 2020*; *Robijns et al., 2022*; *Wilson et al., 2022*). Since the pathophysiology RIF is within therapeutic influence of PBMT at the cellular level, it could potentially aid in reducing fibrosis severity, improving tissue elasticity, and restoring function in affected areas. (*Mamalis, Siegel & Jagdeo, 2016*; *Vallée et al., 2017*; *Tripodi et al., 2021*). The existing evidence base is, however, limited by absence of a structured review and comprehensive evaluation of its efficacy and mechanism of action. As this is an emerging area of research, a structured scoping review is critical to consolidate evidence, identify gaps, and provide a foundation for clinical studies and future research. Therefore, this scoping review was undertaken with the aim of summarizing (i) current state of *in vitro*, *in vivo* research and clinical studies on effectiveness of PBMT in prevention and treatment of RIF; (ii) proposed photobiomodulation induced mechanisms for prevention and treatment of RIF.

## METHODOLOGY

### Registration

The review was prospectively registered on OSF (https://osf.io/ut5fe) on 8th of February 2024.

### Search strategy

A comprehensive search was performed in five databases (PubMed, Scopus, LILACS, ProQuest, EBSCO) on 19th of February 2024 and rerun on 21st September 2024. The terms used for the search included five variations for radiation induced fibrosis and sixteen variations for photobiomodulation. Boolean operators 'OR' was used between variations and 'AND' was used between the terms. The search results from inception to date of search were included for screening. A targeted search also was run on Google Scholar, ResearchGate and Academia.edu and in addition, back references were screened from the relevant articles. The search strategy used for PubMed was: (((((Radiation fibrosis syndrome [MeSH]) OR (Radiation induced fibrosis)) OR (Radiation-induced fibrosis)) OR (Radiation fibrosis syndrome)) OR (Chronic radiation injury)) OR (Radiotherapy fibrosis) AND ((((((((((((((((Photobiomodulation) OR (Low level laser therapy)) OR (Low-level laser therapy)) OR (Low level light therapy)) OR (Low-level light therapy)) OR (Low power light therapy)) OR (Low-power light therapy)) OR (Light-emitting diode)) OR (Light emitting diode)) OR (Red light)) OR (Infrared light)) OR (Phototherapy)) OR (Biostimulation)) OR (PBMT)) OR (LLLT)) OR (LED)) OR (Low-level light therapy [MeSH]). The detailed search strategy of all the databases is mentioned in Data S1.
### Selection of studies

The results of each database were exported to Rayyan Software (*Ouzzani et al., 2016*), compiled and duplicates were removed. Title, abstract, and full-text screening were conducted independently by two reviewers (RG and BT). Studies were evaluated based on predefined eligibility criteria. Disagreements, if any, were resolved through discussions with the senior author (SKV). For a study to be included in the review, it had to be either (i) *in vitro*, *in vivo* or clinical studies that have evaluated the effect of PBMT on RIF; OR (ii) studies that have described or proposed mechanism of action of PBMT on RIF. Reviews and studies reporting the effects of other interventions for RIF were excluded. In addition, we had proposed to exclude studies describing intervention for non-radiation related fibrosis.

### Data extraction and management

Data was extracted by RG and BT from the included studies using a structured data extraction proforma. Details of study design and demographic details of participants, photobiomodulation parameters (including wavelength, energy density, and treatment frequency), proposed mechanisms of action reported in the studies, adverse effects observed if any, key findings and outcomes of interest were extracted to Microsoft Excel (*Microsoft Excel Spreadsheet Software |Microsoft 365, 2024*).

### Result table generation

A qualitative synthesis of the included studies was performed and included studies were segregated based on study type as *in vitro*, *in vivo* and clinical studies. Detailed tables summarizing the photobiomodulation parameters, observed outcomes and reported mechanisms were created for each type of studies. This approach provided a structured and comprehensive summary of the findings, enabling clear comparisons and identification of research gaps.

### Assessment of study quality

Although this was not initially planned during the review registration, we conducted a quality assessment of clinical studies. The quality of the clinical studies was evaluated by RG and BT using the NIH Quality Assessment Tool (*NIH, 2023*). Disagreements, if any, were resolved after discussing with SKV.

## RESULTS

### Selection of studies

The comprehensive data search retrieved 2,733 (PubMed-680, Scopus-411, EBSCO-217, ProQuest-926, LILACS-497, other platforms-02) studies. After manual removal of 815 duplicates, 1,918 studies were included for title and abstract screening by RG and BT. Following resolution of disagreement for 11 studies with SKV, 28 were shortlisted for full-text screening of which only two studies (one clinical and *in vivo* study) met the predefined inclusion criteria (*Mosca et al., 2019*; *Paim et al., 2022*). However, several studies had used PBMT for prevention and treatment of fibrosis due to non-radiation etiology. Among the 23 such studies, nine were *in vitro* studies that evaluated effect of PBMT on fibroblast

(*Webb, Dyson & Lewis, 1998*; *Mamalis, Garcha & Jagdeo, 2015*; *Sassoli et al., 2016*; *Mamalis et al., 2016*; *Yeh et al., 2017*; *Mignon et al., 2018*; *Lee et al., 2020*; *Lee et al., 2021*; *Austin et al., 2021*); 12 were *in vivo* studies in which fibrosis were either induced chemically (*Alessi Pissulin et al., 2017*; *Chiang et al., 2020*; *Gonçalves et al., 2023*), or by cryolesion (*Mesquita-Ferrari et al., 2011*; *De Souza et al., 2011*; *Assis et al., 2013*; *França et al., 2013*; *Alves et al., 2014*), or by contusion (*Luo et al., 2013*), or in animal models that mimicked muscle fibrotic changes in Duchenne muscular dystrophy (DMD) (*Leal-Junior et al., 2014*; *Tomazoni et al., 2020*; *Covatti et al., 2024*); and two were clinical studies that included patients with oral submucous fibrosis (OSMF) (*Chandra, Gujjari & Sankar, 2019*; *Sukanya et al., 2022*). Although fibrosis in these studies lacked the repeated radiation exposure, they share overlapping mechanisms with RIF such as inflammation, oxidative stress, and tissue remodeling, that are targeted by PBMT. Considering the paucity of literature using radiation exposure among *in vitro* and *in vivo* studies, and the wealth of information these 23 studies could add to this review's objectives, we decided to include them. Additionally, one study was identified that utilized high-level laser therapy (HLLT) for the treatment of RIF (*Wilson et al., 2023*) and included in the review. Though HLLT is generally known to deposit higher energy for a given treatment time, the amount of energy deposited to the target tissue can be modulated by adjusting the irradiance and treatment time to achieve similar energy deposition as PBMT to achieve similar therapeutic benefits (*Basalamah et al., 2013*; *Lu et al., 2021*; *Liu et al., 2023*). Thus, a total of 26 studies (two studies using PBMT on RIF, 23 studies using PBMT on fibrosis due to non-radiation etiology and one study using HLLT on RIF) were included in the review to provide relevant insights into the effects and therapeutic mechanisms of PBMT for managing fibrosis. The details of the screening process are summarized using the PRISMA flowchart in Fig. 1. In addition, for greater clarity; type of included studies, etiology of fibrosis, and treatment modality is schematically represented using Venn diagram in Fig. 2.

## Type of included studies

The included studies comprised of nine *in vitro* studies, 13 *in vivo* studies, and four clinical studies. The study characteristics, interventions and the effect of interventions have been categorized and described based on their respective study types in the further sections and is summarized in Table 1.

## Quality assessment of human studies

Three out of four clinical studies were included for quality analysis. One study was a case report and not considered for quality analysis. Among the three studies, the appropriate checklist of NIH Quality Assessment tool was used based on the study design. Two studies (one pre-post design (*Sukanya et al., 2022*) and the other a case series (*Paim et al., 2022*)) were rated as good quality and one case series (*Wilson et al., 2023*) was rated as fair quality. The summary of the NIH quality assessment is represented in Table 2.
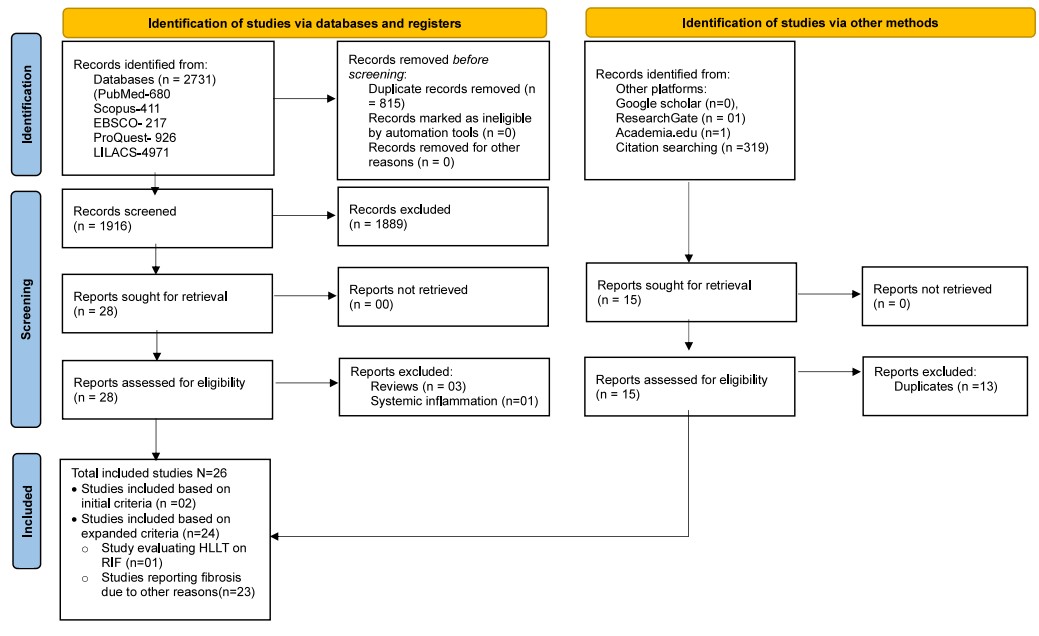

**Figure 1  PRISMA flow diagram.**

## Characteristics and intervention details of included studies
### *In vitro studies*

Nine of the 26 studies included *in vitro* cell lines of which all of them studied the effect of PBMT on fibrosis induced due to the reasons other than radiation exposure. The studies evaluated the effect of PBMT on fibroblasts derived from murine embryo (*Sassoli et al., 2016*; *Lee et al., 2020*; *Lee et al., 2021*) or human dermal (*Webb, Dyson & Lewis, 1998*; *Mamalis, Garcha & Jagdeo, 2015*; *Mamalis et al., 2016*; *Mignon et al., 2018*; *Austin et al., 2021*) or gingival tissue (*Yeh et al., 2017*). All the *in vitro* studies aimed at prevention of fibrosis. Five of the nine studies mentioned the type laser source to be GaAlAs (*Mamalis, Garcha & Jagdeo, 2015*; *Mamalis et al., 2016*; *Yeh et al., 2017*; *Mignon et al., 2018*; *Austin et al., 2021*). The dosage varied between studies with wavelength between 415–633 nm, fluence 0–640 J/cm$^2$, irradiance 0–200 mW/cm$^2$ delivered between 2–25 days.

### *In vivo studies*

Thirteen of the included studies evaluated the effect of PBMT on fibrosis in animal models. In one study, fibrosis was induced by radiation (brachytherapy) (*Mosca et al., 2019*) while the 12 other studies induced fibrosis by other mechanisms (*Mesquita-Ferrari et al., 2011*; *De Souza et al., 2011*; *Assis et al., 2013*; *Luo et al., 2013*; *França et al., 2013*; *Alves et al., 2014*; *Leal-Junior et al., 2014*; *Alessi Pissulin et al., 2017*; *Tomazoni et al., 2020*; *Chiang et al., 2020*; *Gonçalves et al., 2023*; *Covatti et al., 2024*). The mechanisms of induction of fibrosis included chemically (bleomycin (*Chiang et al., 2020*), Bupivacaine (*Alessi Pissulin et al., 2017*) Ketamine and Xylazine (*Gonçalves et al., 2023*)) or by cryolesion (*Mesquita-Ferrari et al., 2011*; *De Souza et al., 2011*; *Assis et al., 2013*; *França et al., 2013*; *Alves et al., 2014*), or by contusion (*Luo et al., 2013*), or in animal models that mimicked muscle fibrotic

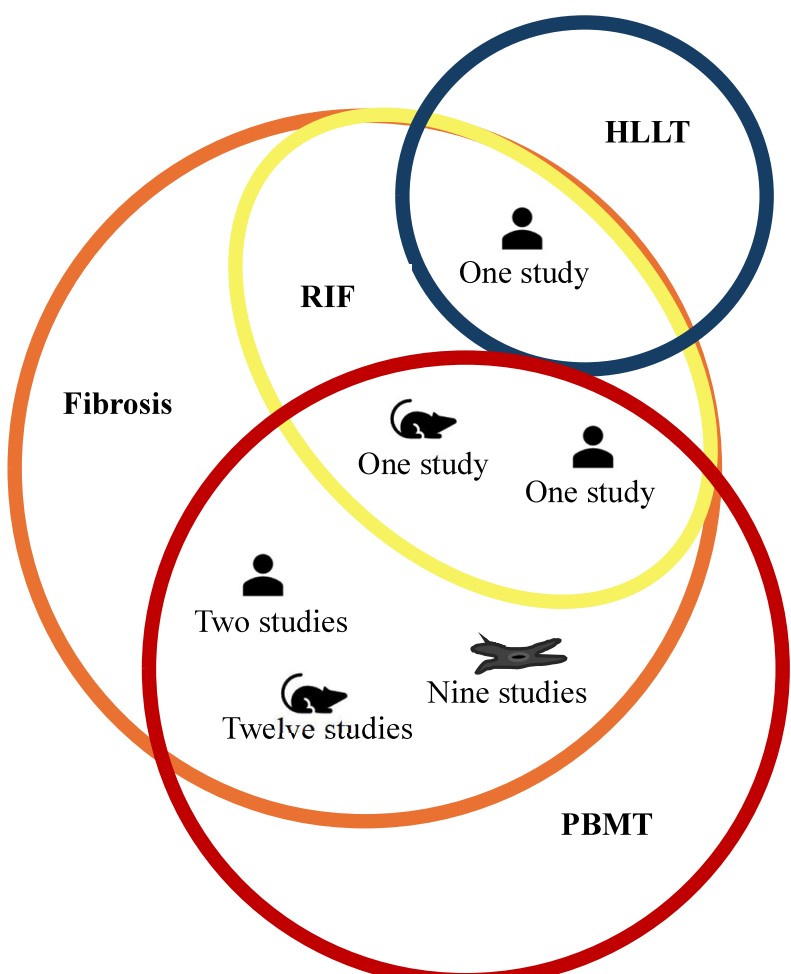

**Figure 2** **Schematic representation of PBMT usage across types of studies.** The fibrosis etiology, intervention and the number of *in vitro* studies (represented as fibroblast cell icon), *in vivo* studies (represented as a rodent icon) and clinical trials (represented as person icon). The orange circle represents fibrosis with radiation induced fibrosis (RIF) (represented as yellow circle) being the subset of fibrosis. The blue colored circle represents high level laser therapy (HLLT) while red colored circle represents photobiomodulation (PBMT).

changes in Duchenne muscular dystrophy (DMD) (*Leal-Junior et al., 2014*; *Tomazoni et al., 2020*; *Covatti et al., 2024*). The *in vivo* studies were conducted on either rats ($n = 433$ across eight studies) or mice ($n = 203$ across five studies). All the studies were randomized controlled trials (RCTs), with two focusing on the treatment of fibrosis (*Tomazoni et al., 2020*; *Covatti et al., 2024*) and the other 11 aimed at its prevention (*Mesquita-Ferrari et al., 2011*; *De Souza et al., 2011*; *Assis et al., 2013*; *Luo et al., 2013*; *França et al., 2013*; *Alves et al., 2014*; *Leal-Junior et al., 2014*; *Alessi Pissulin et al., 2017*; *Mosca et al., 2019*; *Chiang et al., 2020*; *Gonçalves et al., 2023*). Among the included studies, one aimed at prevention of RIF, 10 and one study aimed at prevention and treatment of fibrosis due to non-radiation etiologies, respectively. The source of PBMT included Gallium Arsenide (GaAs)

(*Leal-Junior et al., 2014*; *Alessi Pissulin et al., 2017*), aluminium gallium indium phosphide (AlGaAlP) (*Mesquita-Ferrari et al., 2011*; *De Souza et al., 2011*) and gallium aluminum arsenide (GaAlAs) (*Assis et al., 2013*; *Luo et al., 2013*; *França et al., 2013*; *Alves et al., 2014*; *Chiang et al., 2020*; *Gonçalves et al., 2023*). The parameters of PBMT varied between studies (wavelength: 635–904 nm, fluence: 5–180 J/cm$^2$, irradiance: 0.5–500 mW/cm$^2$). The study that evaluated RIF (*Mosca et al., 2019*) used the following dosage: irradiance—40 mW/cm$^2$, fluence—20 J/cm$^2$, mode—continuous, duration—60 days.

### Clinical studies

Among the four included studies, two included participants with RIF (*Paim et al., 2022*; *Wilson et al., 2023*) while the other two (*Chandra, Gujjari & Sankar, 2019*; *Sukanya et al., 2022*) included participants with OSMF. *Paim et al. (2022)* evaluated the effect of PBMT on RIF on six participants, between three to fifteen months after completion of external beam radiotherapy for squamous cell carcinoma of oral cavity or oropharynx. *Wilson et al. (2023)* evaluated the use of HLLT on RIF in five participants between three months to 40 years post external beam radiotherapy among survivors of head and neck, breast or reticulum cell carcinoma. A single case report by *Chandra, Gujjari & Sankar (2019)* and a pre-post intervention study on 30 participants by *Sukanya et al. (2022)* evaluated the effects of PBMT on OSMF. These studies targeted treatment of RIF (*Paim et al., 2022*; *Wilson et al., 2023*) or fibrosis due to non-radiation etiology (*Chandra, Gujjari & Sankar, 2019*; *Sukanya et al., 2022*). *Paim et al. (2022)* used PBMT (wavelength: 660 nm (red) and 808 nm (IR), spot size: 3.3 mm$^2$, energy: 6 J/point, fluence: 199.98 J/cm$^2$) to treat RIF on their six participants, whereas *Wilson et al. (2023)* used HLLT (wavelength: 532–596 nm, spot size: 5–10 mm, fluence: 5.6 J/cm$^2$) to treat RIF in their five participants. For their participants with OSMF in their respective studies, *Chandra, Gujjari & Sankar (2019)* and *Sukanya et al. (2022)* used PBMT in the range of 808–830 nm, 0.1–0.8 W, four cycles of 15 s each.

## Effect of intervention in included studies
### Safety profile

None of the studies reported any adverse effects due to PBMT or HLLT delivery during the intervention phase or follow-up where applicable.

The *in vitro* studies reported inhibition of alpha smooth muscle actin ($\alpha$-SMA), transforming growth factor-beta (TGF-$\beta$1), AKT/PI3k, collagen production, fibroblast proliferation and increased matrix metalloproteinases-1 (MMP-1) expression and cell counts (*Webb, Dyson & Lewis, 1998*; *Mamalis, Garcha & Jagdeo, 2015*; *Sassoli et al., 2016*; *Mamalis et al., 2016*; *Yeh et al., 2017*; *Mignon et al., 2018*; *Lee et al., 2020*; *Lee et al., 2021*; *Austin et al., 2021*). One notable finding was that the effects of PBMT on cellular processes were dose dependent: lower doses stimulating (*Webb, Dyson & Lewis, 1998*) and doses >30 J/cm$^2$ inhibiting cell counts (*Mignon et al., 2018*).

The *in vivo* studies showed reduced IgG uptake, macrophage infiltration, TGF-$\beta$1, $\alpha$-SMA, tumor necrosis factor-alpha (TNF-$\alpha$), creatine kinase (CK), malondialdehyde (MDA), connective tissue growth factor (CTGF), connective tissue thickening and increased Vascular Endothelial Growth Factor (VEGF) expression, angiogenesis, MyoD and

Gururaj et al. (2025), *PeerJ*, DOI 10.7717/peerj.19494

**Table 1   Summary of study characteristics, PBMT parameters and their effects.**

*In vitro* studies

| Author, year | Prevention/ treatment | Cell type and line | Fibrosis induction type | Sample size and groups | PBMT source | Dosage | Duration | Results |
|---|---|---|---|---|---|---|---|---|
| *Webb, Dyson & Lewis (1998)* | Prevention | Human dermal fibroblasts | N/A | NS | Noncoherent Omegasuperluminous diode; | Wavelength: 660 nm, Fluence: 2.4 and 4 J/cm$^2$, Irradiance: 17 mW/cm$^2$, Power: NP, Time: NP | 1 day | Increased cell counts compared to controls. |
| *Mamalis, Garcha & Jagdeo (2015)* | Prevention | Human dermal fibroblasts | N/A | NS | GaAlAs | Wavelength: 415 nm, Fluence: 5–80 J/cm$^2$, Irradiance: 35 mW/cm$^2$, Power: NP, Time: NP | 2 days | Decreased proliferation and increased ROS in a dose-dependent manner. |
| *Sassoli et al. (2016)* | Prevention | Murine embryonic fibroblasts-NIH/3T3 | N/A | NS | Diode laser | Wavelength: 635 nm, Fluence: 0.3 J/cm$^2$, irradiance: NP, Power: 89 mW, Time: NP, Continuous Mode | 3 days | Inhibited TGF-β1-induced fibroblast/myofibroblast transition, upregulated MMP-2, MMP-9, downregulated TIMP-1, TIMP-2. |
| *Mamalis et al. (2016)* | Prevention | Human dermal fibroblasts | N/A | NS | GaAlAs | Wavelength: 633 nm, Fluence: 80, 160, 320, 640 J/cm$^2$, Irradiance: 87 mW/cm$^2$, Power: NP, Time: NP | 2 days | Inhibited collagen production and fibroblast proliferation, increased ROS, inhibited AKT/PI3k. |

Gururaj et al. (2025), *PeerJ*, DOI 10.7717/peerj.19494

**Table 1** (*continued*)

| | | | | | | | | |
|---|---|---|---|---|---|---|---|---|
| *Yeh et al. (2017)* | Prevention | Human healthy marginal Gingival tissue | N/A | NS | GaAlAs | Wavelength: NP, Irradiance: 15.17 mW/cm$^2$, Fluence: 8 J/cm$^2$, Power: NP, Time: NP | 5 days | Reduced CCN2 and $\alpha$-SMA in PBM group. |
| *Mignon et al. (2018)* | Prevention | Human dermal fibroblasts | N/A | NS | GaAlAs | Wavelength: 450, 490, 530 nm Fluence: 0–250 J/cm$^2$, Irradiance: 0–100 mW/cm$^2$, Power: NP, Time: NP | 1 day | Inhibited collagen production and fibroblast proliferation. Increased ROS. Inhibited TGF-$\beta$2. Cytotoxic if >30 J/cm$^2$. |
| *Lee et al. (2020)* | Prevention | NIH/3T3, Murine embryonic fibroblasts | N/A | NS | CNI laser+ PHL assisted | Wavelength: 635 nm, Fluence: 0.3–3 J/cm$^2$, Irradiance: 10–100 mW/cm$^2$, Power: 25–200 mW, Time: 30 s | 2 days | Decreased $\alpha$-SMA, TGF-$\beta$1, and type I collagen in PHL+LLLT group. |
| *Lee et al. (2021)* | Prevention | NIH/3T3, Murine embryonic fibroblasts | N/A | NS | CNI laser+ PHL assisted | Wavelength: 635 nm, Fluence: 8 J/cm$^2$, Irradiance: NP, Power: NP, Time: NP | 21 days | Increased anti-inflammatory effect by 36%; reduced type I collagen, $\alpha$-SMA, and TGF-$\beta$1. |
| *Austin et al. (2021)* | Prevention | Human Dermal fibroblast | N/A | NS | GaAlAs | Wavelength: 633 $\pm$ 30 nm, Fluence: 320 J/cm$^2$ or 640 J/cm$^2$ Irradiance: NP, Power: NP Time: 3,667 s for 320 J/cm$^2$ and 7,334 s for 640 J/cm$^2$ of RL at $\sim$34 °C, | 1 day | RL phototherapy increased MMP-1 expression, enhancing extracellular collagen remodeling. Upregulated PRSS35 with anti-fibrotic functions |

**Table 1** (*continued*)

***In vivo* studies**

| Author, year | Prevention/ treatment | Animal model | Fibrosis induction type | Sample size and groups | PBMT source | Dosage | Duration | Results |
|---|---|---|---|---|---|---|---|---|
| De Souza et al. (2011) | Prevention | Wistar rats | Cryolesion | $n = 5$ (Control = 1, Sham = 1, Cryoinjury = 1, Laser-treated cryoinjury = 2) | InGaAlP | Wavelength: 660 nm, Fluence: 5 J/cm$^2$, Irradiance: 0.5 mW/cm$^2$, Power: 20 mW, Exposure: 10 s, Total energy: 0.2J, Beam spot: 0.04 cm$^2$ | 7 days | Reduced myonecrosis and increased angiogenesis in the laser-treated group. Collagen types I and III deposition significantly increased on day 7. |
| Assis et al. (2013) | Prevention | Wistar rats, tibialis anterior muscle | Cryolesion | $n = 60$ (Control group = 20, Injured TA = 20, Injured TA+LLLT = 20) | AlGaAs | Wavelength: 808 nm, Fluence: 180 J/cm$^2$, Irradiance: 3,800 mW/cm$^2$, Power: 30 mW, Time: NP, Energy: 1.4J | 4 days | LLLT decreased lesion percentage area, increased MyoD and Myogenin mRNA, reduced TGF-β1, and improved VEGF expression. |
| Alves et al. (2014) | Prevention | Wistar rats, tibialis anterior | Cryolesion | $n = 110$ (Control = 10, Sham = 10, LLLT = 30, Non-Treated Injury = 30, Injury+LLLT = 30) | AlGaAs | Wavelength: 780 nm, Fluence: 10 J/cm$^2$, Irradiance: 1,000 mW/cm$^2$, Output Power: 40 mW, Exposure Time: 10 s, Beam Spot: 0.04 cm$^2$ | 7 days | LLLT reduced inflammatory infiltrate and myonecrosis (Day 1), increased blood vessels (Days 3 & 7), and increased immature muscle fibers and MMP-2 activity. |

**Table 1** (*continued*)

| | | | | | | | | |
|---|---|---|---|---|---|---|---|---|
| *Mesquita-Ferrari et al. (2011)* | Prevention | Wistar rat | Cryolesion | $n = 35$ (Untreated = 5, Cryo Injury = 15, Cryo+LLLT = 15) | AlGaInP laser | Wavelength: 660 nm, Fluence: 5 J/cm$^2$, Irradiance: 500 mW/cm$^2$, Power: 20 mW, Time: 10 s Beam spot: 0.04 cm$^2$, | 14 days | Reduced TNF-$\alpha$ and TGF-$\beta$ levels. |
| *França et al. (2013)* | Prevention | Wistar tat, diabetic | Cryolesion | $n = 65$ (SHAM = 2, Control = 5, Diabetic = 5, SHAM (Diabetic) = 5, LLLT = 15, D-LLLT = 15, $D = 15$) | GaAlAs | Wavelength: 750 nm, Fluence: 5 J/cm$^2$, Irradiance: 500 mW/cm$^2$, Power: NP, Time: 10 s/point | 14 Days | Accelerated remodeling phase in LLLT group, while diabetic group remained in proliferative fibrosis phase. |
| *Luo et al. (2013)* | Prevention | Sprague-dawley rats, gastronemius muscle | Contusion | $n = 96$ (No lesion untreated = 6, Contusion = 48, Contusion +LLLT = 42) | GaAlAs | Wavelength: 635 nm, Fluence: 21 J/cm$^2$, Irradiance: 17.5 mW/cm$^2$, Power: 7 mW, Time: 20 min, Beam spot: 0.4 cm$^2$, | 28 days | LLLT increased IGF-1 and SOD activity, reduced MDA levels in the first week, and later decreased IGF-1 and TGF-$\beta$1 |
| *Leal-Junior et al. (2014)* | Prevention | Mdx mice, tibialis anterior | DMD | $n = 10$ Superpulsed LLLT = 5 Placebo LLLT = 5 | GaAs | Wavelength: 904 nm, Fluence: NP, Irradiance: NP, Power: 15 mW, Time: NP, Frequency: 700 Hz, Energy: 1J | 14 weeks | Reduced muscle atrophy and fibrosis, lower CK levels, and significantly decreased inflammatory markers (*e.g.*, TNF-$\alpha$, IL-1$\beta$, IL-10, COX-2) with LLLT. |

Gururaj et al. (2025), *PeerJ*, DOI 10.7717/peerj.19494

**Table 1** (*continued*)

| | | | | | | | | |
|---|---|---|---|---|---|---|---|---|
| *Alessi Pissulin et al. (2017)* | Prevention | Wistar rats, sternocleido-mastoid muscle | Bupivacaine | n = 30 (Control group = 15, Laser group = 15) | GaAs | Wavelength: 904 nm, Fluence: NP, Irradiance: NP, Power: 50 mW, Time: NP, Energy: 2.8 J/point, | Treated for 5 days, assessed on day 12 | LLLT reduced fibrosis, myonecrosis, and CK levels. |
| *Tomazoni et al. (2020)* | Treatment | MDX Mice | DMD | n = 90 (Wildtype = 5, Placebo Control = 10, PBMT = 15, Prednisone = 15, NSAID = 15, PBMT+Prednisone = 15, PBMT+NSAID = 15) | LED Diode | Cluster probe with 9 diodes (1 laser: Wavelength: 905 nm, 4 LEDs: wavelength: 875 nm, 4 LEDs: wavelength: 640 nm), Fluence: NP, Irradiance: NP, Power: NP, Time: NP | 3x/week for 14 weeks | Prednisone + PBMT (alone or combined) preserved muscle morphology and improved functional performance |
| *Chiang et al. (2020)* | Prevention | BALB/c Mice | Bleomycin | n = 46 (PBS = 12, BLM = 12, ANE = 12, ANE+PBM = 5, ANE+Forskolin = 5) | GaAlAs | Wavelength: 660 nm, Fluence: 8 $J/cm^2$, Irradiance: 15.17 $mW/cm^2$, Power: NP, Time: NP | 30 days | Reduced $\alpha$-SMA and CTGF. |
| *Gonçalves et al. (2023)* | Prevention | Wistar rats with SMA, gastrocnemius muscle | Ketamine and xylazine + immobilisation for 5days | n = 32 (Control group = 8, Immobilized control = 8, Immobilized+Red Laser = 8, Immobilized+IR Laser = 8) | GaAlAs | Wavelength: 660 nm or 808 nm, Fluence: 60 $J/cm^2$, Irradiance: 1,070 $mW/cm^2$, Power: 30 mW, Time: 56 s, Spot Area: 0.028 $cm^2$, Continuous mode | 9 days | Reduced inflammatory infiltrate and connective tissue thickening; IR laser showed muscle fiber regeneration and increased oxidative fibers (type I). |

**Table 1** (*continued*)

| Author, year | Prevention/treatment | Population | Fibrosis induction type | Sample size and groups | PBMT source | Dosage | Duration | Results |
|---|---|---|---|---|---|---|---|---|
| *Mosca et al. (2019)* | Prevention | Athymic mice | Brachytherapy (RIF) | $n = 36$: Control (6), Red Laser (6), NIR Laser (6), RT (6) RT + Red Laser (6), RT + NIR (6) | NS | Wavelength: NP, Fluence: 20 J/cm$^2$, Irradiance: 40 mW/cm$^2$, Power: NP, Time: NP, Continuous wave | 60 Days | Less temperature (inflammation) and normal morphology of tissues and lesser thickening, better vascular perfusion in PBMT groups (NIR<red) |
| *Covatti et al. (2024)* | Treatment | MDX mice | DMD | $n = 21$ (Untreated Wild Type = 7, Untreated MDX = 7, Treated MDX = 7) | NS | Wavelegnth: NP, Fluence: NP, Irradiance: NP, Power: NP, Time: NP, Energy: 0.6J | 3x/week for 42 days | Reduced IgG uptake, macrophage infiltration, and improved histomorphology features. |

**Clinical studies**

| Author, year | Prevention/treatment | Population | Fibrosis induction type | Sample size and groups | PBMT source | Dosage | Duration | Results |
|---|---|---|---|---|---|---|---|---|
| *Chandra, Gujjari & Sankar (2019)* | Treatment | Patient with oral submucous fibrosis | Oral submucous fibrosis (non-RT related) | $n = 1$ | Diode laser | Wavelength: 808 nm using 600 nm optic fiberin, Fluence: NP, Irradiance: NP, Time: 10 s at power: 800 mW in 3 cycles, continuous mode | Treatment: 3 day, follow up: 30 days | Improvement in mouth opening 10 mm |
| *Sukanya et al. (2022)* | Treatment | Patients with oral submucous fibrosis | Oral submucous fibrosis (non-RT related) | $n = 30$ | BTL-5,000 series | Wavelength: 830 nm, Fluence: NP, Irradiance: NP, Power: 100 mW, 4 cycles of time: 15 s each on Days 0, 3, 7, 15 | Follow up at 1, 3 and 6 months | LLLT improved mouth opening (Day 0 to 15: 9.91 ± 3.34 mm; Day 1 to 6 months: 14.29 ± 6.82 mm). |

Gururaj et al. (2025), *PeerJ*, DOI 10.7717/peerj.19494

**Table 1** (*continued*)

| | | | | | | | | |
|---|---|---|---|---|---|---|---|---|
| *Paim et al. (2022)* | Treatment | SCC of Oral cavity Received RT 3–15 months prior | External beam RT (RIF) | $n = 6$ (OMT = 3, OMT+PBMT = 3) | GaA1As and InGaAlP | Wavelength: 660 nm and 808 nm, Fluence: 199.98 J/cm$^2$, Irradiance: NP, Power: NP, Time: NP, Spot size: 3.3 m$^2$, Energy: 6 J/point, Continuous mode | 5 weeks | OMT increased mouth opening by 9.25 mm; OMT+PBMT increased by 23.1 mm with better tolerance and reduced pain. |
| *Wilson et al. (2022)* | Treatment | 3 HNC, 1 Breast, 1 Reticulum Cell Sarcoma | External beam RT (RIF) | $n = 5$ | KTP, PDL, CO$_2$ Laser | Wavelength: 532–596 nm, Fluence: 5.6 J/cm$^2$, Irradiance: NP Power: NP, Time: NP, Spot size: 5–10 mm | Treatment: 3 to 12 days follow up:4–24 weeks | Reduced pain, scarring, discoloration, and improved range of motion. |

**Notes.**

$\alpha$-SMA, Alpha Smooth Muscle Actin; CCN, Cellular Communication Network factor; COX-2, Cyclooxygenase-2; CTGF, Connective Tissue Growth Factor; DMD, Duchenne Muscular Dystrophy; GaAs, Gallium Arsenide; GaAlAs, Gallium Aluminum Arsenide; IgG, Immunoglobulin G; IL-1$\beta$, Interleukin-1 Beta; IL-10, Interleukin-10; InGaAlP, Indium Gallium Aluminum Phosphide; KTP, Potassium-Titanyl-Phosphate Lase; LED, Light Emitting Diode; LLLT, Low-Level Laser Therapy; MDX, Mouse Model for Duchenne Muscular Dystrophy; MMP, Matrix Metalloproteinases; MyoD, Myogenic Differentiation Factor D; N/A, Not Applicable; NIR, Near-Infrared; NP, Not Provided; NS, Non-Specified; OMT, Osteopathic Manipulative Therapy; PBMT, Photobiomodulation Therapy; PDL, Pulsed Dye Laser; PGL, Phloroglucinol; SMA, Spinal Muscular Atrophy; TGF-$\beta$, Transforming Growth Factor-Beta; TIMP, Tissue Inhibitor of Metalloproteinases; TNF-$\alpha$, Tumor Necrosis Factor-Alpha; VEGF, Vascular Endothelial Growth Factor.
**Table 2  Summary of assessment of study quality using NIH quality assessment tool.**

| Author, year | Type of study | Item numbers | | | | | | | | | | | | Overall score |
|---|---|---|---|---|---|---|---|---|---|---|---|---|---|---|
| | | 1 | 2 | 3 | 4 | 5 | 6 | 7 | 8 | 9 | 10 | 11 | 12 | |
| *Sukanya et al. (2022)* | Pre-post study | Yes | Yes | Yes | Yes | Yes | Yes | Yes | No | Yes | Yes | Yes | NA | Good |
| *Paim et al. (2022)* | Case series | Yes | Yes | No | No | Yes | Yes | Yes | Yes | Yes | – | – | – | Good |
| *Wilson et al. (2022)* | Case series | Yes | Yes | No | No | Yes | No | Yes | No | Yes | – | – | – | Fair |

Notes.
  NA, not applicable.
  Overall score: ≥75% of the maximum score- good, <75%–50% of maximum score- fair and <50% of maximum score- poor quality.

myogenin mRNA. Notable findings were that with PBMT exposure, collagen deposition increased during the acute phase, and there was a faster trigger of remodeling phase (*Mesquita-Ferrari et al., 2011*; *De Souza et al., 2011*; *Assis et al., 2013*; *Luo et al., 2013*; *França et al., 2013*; *Alves et al., 2014*; *Leal-Junior et al., 2014*; *Alessi Pissulin et al., 2017*; *Mosca et al., 2019*; *Tomazoni et al., 2020*; *Chiang et al., 2020*; *Gonçalves et al., 2023*; *Covatti et al., 2024*). In addition, IGF-1 showed to increase in the first week and gradually tapered by the 28th day (*Luo et al., 2013*). These findings highlight the healthy wound healing regulation by PBMT. In the MDX mice model for DMD, prednisone with PBMT showed an additive effect on treatment of fibrosis by improving muscle morphology and functional outcomes (*Tomazoni et al., 2020*) and addition of phloroglucinol with PBMT further improved anti-inflammatory effect (*Lee et al., 2020*; *Lee et al., 2021*).

The clinical studies focused on evaluation of clinical and functional outcomes rather than cellular response to PBMT. They demonstrated reduction in pain, scarring and discoloration as well as improvements in range of motion immediately after treatment as well as during the follow-up period (*Chandra, Gujjari & Sankar, 2019*; *Sukanya et al., 2022*; *Paim et al., 2022*; *Wilson et al., 2023*). This demonstrated the long-term maintenance of its effects on functional outcomes.

## Reported mechanism of action of PBMT on mitigation of fibrosis

The 20 studies that investigated the use of PBMT for the prevention of fibrosis, including one specifically for RIF, collectively highlighted several key mechanisms by which PBMT mitigates the development of fibrotic tissue. The primary mechanism involved the inhibition of the early steps in the transition of fibroblasts to myofibroblasts, modulation of ECM and collagen production (*De Souza et al., 2011*; *Assis et al., 2013*; *Alves et al., 2014*; *Mamalis, Garcha & Jagdeo, 2015*; *Sassoli et al., 2016*; *Alessi Pissulin et al., 2017*; *Mignon et al., 2018*; *Mosca et al., 2019*; *Lee et al., 2020*; *Lee et al., 2021*; *Tomazoni et al., 2020*; *Austin et al., 2021*). In addition to its effects on fibroblast activity, PBMT reduced pro-inflammatory and profibrotic signaling pathways (*Webb, Dyson & Lewis, 1998*; *Mesquita-Ferrari et al., 2011*; *Assis et al., 2013*; *Mamalis, Garcha & Jagdeo, 2015*; *Mamalis et al., 2016*; *Lee et al., 2020*; *Lee et al., 2021*; *Tomazoni et al., 2020*). Fibrosis is often characterized by chronic inflammation, where an overactive immune response leads to the persistent release of cytokines and growth factors that stimulate fibroblast activity. By modulating the inflammatory response, PBMT not only prevented the activation of fibroblasts but also

mitigates the overall fibrotic environment (*Mesquita-Ferrari et al., 2011*; *Leal-Junior et al., 2014*; *Mignon et al., 2018*). Furthermore, PBMT enhanced angiogenesis and improved tissue oxygenation, thereby reducing hypoxia in tissues (*De Souza et al., 2011*; *Assis et al., 2013*; *Alves et al., 2014*). PBMT also modulated oxidative stress by reducing reactive oxygen species (ROS), which are known to damage cellular structures and exacerbate inflammation and fibrosis (*Luo et al., 2013*; *Mamalis, Garcha & Jagdeo, 2015*; *Mamalis et al., 2016*). PBMT also triggered the remodeling phase of wound healing and improved tissue architecture (*França et al., 2013*; *Gonçalves et al., 2023*). Collectively, these mechanisms contributed to a reduction in the initial stages of fibrosis and improvement of overall tissue health.

The five studies that evaluated PBMT for the treatment of fibrosis, including two focusing on RIF, reported specific mechanisms of action. The studies demonstrated that PBMT caused reduction in fibrosis by triggering the anti-inflammatory processes and the remodeling phase of wound healing; leading to mitigation of fibrotic markers such as excessive collagen deposition and abnormal extracellular matrix remodeling reflected by better muscle histomorphology (*Chiang et al., 2020*; *Covatti et al., 2024*). The regenerative capacity of PBMT was also highlighted by better functional performance and lesser fatigue.

A summary of reported mechanisms of action of PBMT in mitigating fibrosis development and treatment is presented in Table 3.

## DISCUSSION

RIF presents as functional limitations that hinder daily activities, contribute to emotional distress, reduce self-confidence and social engagement, ultimately impairing quality of life during survivorship (*Ramia et al., 2022*; *Wilson et al., 2022*). In addition, the progressive nature of RIF increases the disability over time, making it an important complication to prevent and address in cancer survivors. The complex pathophysiology of RIF, characterized by persistent inflammation, excessive collagen deposition, and progressive tissue stiffness, poses challenges for effective management (*Ejaz, Greenberger & Rubin, 2019*; *Fijardo et al., 2024*). Current prevention and treatment options are limited for this disabling complication and are predominantly in the research phase with mixed clinical outcomes (*Fijardo et al., 2024*). In this context, PBMT has been advocated as a potential therapeutic option for both prevention and treatment of RIF. However, the current advocacy for PBMT is based on weak evidence and limited studies (*Tam et al., 2020*; *Robijns et al., 2022*). This scoping review was warranted to bridge this evidence gap and was executed by a comprehensive literature search using a structured strategy across five databases, two academic social network platforms, Google Scholar and back references of relevant literature.

This scoping review explored the potential of PBMT as a preventive and therapeutic intervention for RIF. Firstly, none of the included studies reported adverse effects, reinforcing the safety of PBMT in cancer populations which is in line with the existing evidence (*De Pauli Paglioni et al., 2019*; *Bensadoun et al., 2020*). In addition, the findings of this review highlight the existence of huge body of evidence from pre-clinical studies on fibrosis, indicating that PBMT may be beneficial in addressing the multifactorial pathophysiology of RIF at a cellular level. The included clinical studies though low on

**Table 3** Summary of reported mechanism of action of PBMT on mitigation of fibrosis.

| Author, year | Prevention/ treatment | Mechanism of action |
|---|---|---|
| *In vitro* **studies** | | |
| *Webb, Dyson & Lewis (1998)* | Prevention | Balanced fibroblast proliferation |
| *Mamalis, Garcha & Jagdeo (2015)* | Prevention | Modulation of fibrotic markers, fibroblast proliferation and oxidative stress |
| *Sassoli et al. (2016)* | Prevention | Inhibition of fibroblast-to-myofibroblast transition, modulation of collagen production |
| *Mamalis et al. (2016)* | Prevention | Modulation of pro-fibrotic markers, fibroblast proliferation and oxidative stress |
| *Yeh et al. (2017)* | Prevention | Reduction in profibrotic factors |
| *Mignon et al. (2018)* | Prevention | Modulation of Collagen Production and inflammatory response |
| *Lee et al. (2020)* | Prevention | Anti-inflammatory effects, modulation of collagen production, pro-fibrotic factors |
| *Lee et al. (2021)* | Prevention | Anti-inflammatory effects, modulation of collagen production, pro-fibrotic factors |
| *Austin et al. (2021)* | Prevention | Gene expression modulation of collagen production |
| *In vivo* **studies** | | |
| *De Souza et al. (2011)* | Prevention | Modulation of collagen production, vascular and angiogenic effects |
| *Assis et al. (2013)* | Prevention | Modulation of collagen production, profibrotic factors, vascular and angiogenic effects |
| *Alves et al. (2014)* | Prevention | Modulation of collagen production, vascular and angiogenic effects |
| *Mesquita-Ferrari et al. (2011)* | Prevention | Modulation of early inflammatory response and pro-fibrotic factors |
| *França et al. (2013)* | Prevention | Trigger of remodelling phase of wound healing |
| *Luo et al. (2013)* | Prevention | Oxidative stress modulation |
| *Leal-Junior et al. (2014)* | Prevention | Anti-inflammatory effects |
| *Alessi Pissulin et al. (2017)* | Prevention | Improved tissue architecture and ECM production modulation |
| *Tomazoni et al. (2020)* | Prevention | Inhibition of fibroblast-to-myofibroblast transition and pro fibrotic factors |
| *Chiang et al. (2020)* | Treatment | Improved tissue architecture and muscle morphology |
| *Gonçalves et al. (2023)* | Prevention | Promotion of oxidative muscle fibre regeneration (type I), increasing tissue elasticity |

**Table 3** (*continued*)

| Author, year | Prevention/ treatment | Mechanism of action |
|---|---|---|
| *Mosca et al. (2019)* | Prevention | Modulation of early inflammatory response and collagen production |
| *Covatti et al. (2024)* | Treatment | Reduced inflammatory mediators, improved tissue architecture and muscle morphology |

evidence hierarchy due to their design, highlighted the beneficial effects of PBMT in improving clinical and functional outcomes. The findings from both pre-clinical and clinical studies allude to the potential of PBMT as a non-invasive treatment option to mitigate the symptoms associated with RIF.

The proposed mechanism of action of PBMT provides a strong theoretical basis for its application in RIF and is in line with the narrative reviews and commentaries published in this domain (*Mamalis, Siegel & Jagdeo, 2016*). Proposed mechanisms in the included studies for both the prevention and treatment of RIF suggest that PBMT supports the regulation of normal wound healing pathways across all phases of healing. In the inflammatory phase, PBMT has shown to reduce the inflammatory cytokine levels (IL-6, TNF-$\alpha$, IL-1$\beta$, IL-10, COX-2) which are typically elevated due to radiation exposure (*Leal-Junior et al., 2014*; *Alessi Pissulin et al., 2017*). Additionally, PBMT enhances the activity of superoxide dismutase which helps neutralize ROS (*Luo et al., 2013*). By mitigating oxidative stress, PBMT limits persistent inflammation and TGF-$\beta$1, a key driver of fibrosis formation (*Mesquita-Ferrari et al., 2011*; *Assis et al., 2013*; *Luo et al., 2013*; *Sassoli et al., 2016*; *Alessi Pissulin et al., 2017*; *Lee et al., 2020*; *Lee et al., 2021*). In the proliferative phase, PBMT is shown to modulate processes such as fibroblast to myofibroblast conversion, collagen synthesis and organization of collagen deposition (*De Souza et al., 2011*; *Assis et al., 2013*; *Alves et al., 2014*; *Sassoli et al., 2016*; *Mignon et al., 2018*; *Mosca et al., 2019*; *Lee et al., 2020*; *Lee et al., 2021*; *Tomazoni et al., 2020*; *Austin et al., 2021*). Moreover, PBMT stimulates endothelial cell proliferation and upregulates Vascular Endothelial Growth Factor expression (VEGF), enhancing angiogenesis and improving tissue vascularization (*De Souza et al., 2011*; *Assis et al., 2013*; *Alves et al., 2014*). In the remodeling phase, PBMT has shown to upregulate MMPs and downregulate tissue inhibitor of metalloproteinases (TIMPs) ensuring prevention of excessive collagen deposition and disorganized matrix formation, which are key factors of RIF (*Alves et al., 2014*; *Sassoli et al., 2016*). Though *in vitro* and *in vivo* studies demonstrate favorable mechanisms of action, there is need to be cognizant of the potential interaction between PBMT and radiotherapy sensitivity in normal and cancer cells in human participants receiving fractionated radiation therapy. Emerging evidence shows that PBMT 685 nm wavelength and fluence of 20 J/cm$^2$ improves radiosensitivity in cancer cells by increasing oxidative stress, inducing DNA damage and promoting apoptosis and autophagy (*Djavid et al., 2017*). However, the dosage parameters should be tailored to maximize damage to tumor cells while protecting the normal cells.

The review also highlighted the relevance of PBMT parameters on the mechanisms and effects. Red light, typically in the 600–700 nm range, were particularly effective for

superficial structures like skin due to their optimal penetration and energy absorption. In contrast, near infrared light, with wavelengths of 800–1,000 nm, showed to penetrate deeper into tissues, making it more suitable for addressing fibrosis in muscles and other deeper structures (*Assis et al., 2013*; *Alves et al., 2014*; *Leal-Junior et al., 2014*; *Alessi Pissulin et al., 2017*; *Mosca et al., 2019*). In addition, PBMT's effects are shown to be highly dose-dependent, with lower energy densities demonstrating stimulatory effect on cell count and proliferation. Conversely, higher energy densities showed to have an inhibitory effect on the cells, highlighting the opportunities for modulating dosing to optimize therapeutic outcomes (*Mignon et al., 2018*). This dose-dependent application reinforces the need for tailor-made PBMT parameters based on the affected tissue's depth and the stage of fibrosis. WALT position paper provides PBMT dosing recommendations for management of various side effects of primary cancer therapies including for RIF. WALT position paper recommends a dosage of 2 Einstein delivered using near infrared LED/laser device with a power output range of 10–150 mW/cm$^2$ for both prevention and treatment of RIF (*Robijns et al., 2022*). This recommendation is generic and allows for modulating dose calculation using photon fluence specific to each wavelength. However, these recommendations of dosing parameters for RIF are based on consensus and there is a need for future studies to validate the dosage recommendations.

This review highlights the current evidence base of PBMT as a therapeutic tool for prevention and treatment of RIF. Though there is significant body of evidence on mechanisms, they primarily come from preclinical studies and limits their direct application to clinical practice. However, these findings justify the increased attention PBMT is receiving as a potential therapeutic tool for mitigation of RIF and lay the foundation for prospective clinical trials to validate the efficacy of PBMT in clinical practice. There has been significant increase in research attention towards the use of PBMT as a therapeutic tool for the management of RIF, but such attention is limited to pre-clinical studies, reviews and a handful of clinical studies that are considered low on evidence hierarchy. A screening of registered clinical trials in Clinical Trials Registry Platforms such as ClinicalTrials.gov, ICTRP, CTRI identified one feasibility clinical trial which is yet to open for recruitment (*National Library of Medicine, 2024*). The registered trial aims to estimate the efficacy of PBMT in the treatment of RIF in patients with head and neck cancer.

Despite the growing research interest towards RIF, it remains a significant challenge in clinical practice. In addition to the potential benefits of PBMT in mitigating RIF, current evidence supports the use of pentoxifylline, vitamin E, botox, sodermix, pravastatin, impedance controlled microcurrent therapy and exercises as therapeutic options for managing RIF (*Gururaj et al., 2024*). Additionally, newer radiotherapy techniques, such as stereotactic radiotherapy, intensity-modulated radiation therapy, image-guided radiation therapy and emerging techniques such as flash radiotherapy aim to deliver high doze targeted towards cancer cells and minimize exposure to healthy cells, thereby reducing side effects (*Chen & Kuo, 2017*; *Tang et al., 2024*).

### Strengths and limitations

This scoping review adds significant value to the existing evidence base by being structured, comprehensive in its database coverage as well as in its inclusion of *in vivo*, *in vitro* and clinical studies. Another strength of this review is in its well-organized details of PBMT parameters, their specific effects and mechanisms of action, thus providing a comprehensive overview of status of research using PBMT in management of RIF. However, this review has few limitations. Firstly, the number of studies explicitly investigating PBMT for RIF was limited. Due to the paucity of studies on RIF, fibrosis of other etiological origins was included to provide a broader perspective on the existing evidence and the potential application of PBMT in RIF. This approach was based on the shared mechanisms of fibrosis such as inflammation, oxidative stress, and extracellular matrix dysregulation which suggest some degree of generalizability in mechanisms of action of PBMT. However, important differences in pathophysiology such as primary triggers for inflammatory pathway activation and effects of repeated radiation exposure in RIF, may significantly influence therapeutic outcomes. These differences underscore the need for further clinical trials to confirm the beneficial effects of PBMT in RIF.

Based on the findings of this review, it appears that integrating PBMT in clinical practice as a therapeutic option for mitigation of RIF could potentially reduce its incidence, severity and burden during cancer survivorship. While research is increasing, adherence to standardized evidence-based PBMT protocols and careful monitoring of tissue responses are necessary to ensure safety and efficacy in clinical settings.

## CONCLUSIONS

The findings of this review highlight the potential of PBMT as a non-invasive, safe modality with supporting evidence from preclinical studies regarding the mechanism of action, but high-quality RCTs in human population are essential to confirm its clinical utility in managing RIF.

### Funding

The authors received no funding for this work. Rachita Gururaj is supported by a Doctoral Research Fellowship from M.S. Ramaiah University of Applied Sciences, Bengaluru, India. The funders had no role in study design, data collection and analysis, decision to publish, or preparation of the manuscript.

### Grant Disclosures

The following grant information was disclosed by the authors:
M.S. Ramaiah University of Applied Sciences, Bengaluru, India.

### Competing Interests

Stephen R. Samuel is an Academic Editor for PeerJ.

## Author Contributions

- Rachita Gururaj conceived and designed the experiments, performed the experiments, analyzed the data, prepared figures and/or tables, authored or reviewed drafts of the article, and approved the final draft.
- Betty Thomas performed the experiments, analyzed the data, prepared figures and/or tables, authored or reviewed drafts of the article, and approved the final draft.
- Manur Gururajachar Janaki conceived and designed the experiments, authored or reviewed drafts of the article, and approved the final draft.
- Vinay Martin D'sa Prabhu conceived and designed the experiments, authored or reviewed drafts of the article, and approved the final draft.
- Rakesh Nagaraju conceived and designed the experiments, authored or reviewed drafts of the article, and approved the final draft.
- Stephen Rajan Samuel conceived and designed the experiments, authored or reviewed drafts of the article, and approved the final draft.
- Sundar Kumar Veluswamy conceived and designed the experiments, performed the experiments, analyzed the data, prepared figures and/or tables, authored or reviewed drafts of the article, and approved the final draft.

## Data Availability

This is a systematic review/meta-analysis.

## Supplemental Information

Supplemental information for this article can be found online at http://dx.doi.org/10.7717/peerj.19494#supplemental-information.

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
