# Peer review of "Potential role of photobiomodulation as a prevention and treatment strategy for radiation induced fibrosis: a review of effectiveness and mechanisms"

_PeerJ, doi:10.7717/peerj.19494_

## Round 0.1 · original submission · Minor Revisions

Both reviewers have somewhat minor comments. Please respond to them in an appropriate revision

·

Basic reporting

no comment

Experimental design

no comment

Validity of the findings

no comment

Additional comments

Gururaj et al. review photobiomodulation as a prevention and treatment for radiation-induced fibrosis, a clinically relevant issue affecting cancer survivors' quality of life. The review is well-designed, informative, and methodologically sound. However, only two studies specifically address radiation-induced fibrosis, while most available research provides indirect implications within radiotherapy. The authors might consider adjusting the title to reflect this. Overall, I recommend acceptance with minor revisions:

• Discussion (prevention of fibrosis) from line 316:
The review describes various effects in preventing radiation-induced fibrosis. However, some also impact ionizing radiation’s effect on tumors and normal tissue, influencing radiosensitivity (e.g., angiogenesis, oxygen and ROS status). The authors should discuss potential interactions and consequences of such effects for tumor and normal tissue with fractionated radiotherapy. Since photobiomodulation is locally limited before radiotherapy for prevention, tumor volume can likely be omitted from this discussion.

• The authors should also discuss other potential preventive approaches for radiation-induced fibrosis, such as FLASH radiotherapy, which significantly reduces side effects in normal tissue, including fibrosis.

·

Basic reporting

No comment

Experimental design

The authors are encouraged to report in a more consistent manner for the dosage column in Table 1. Specifically, all entries should be standardized to either power density (irradiance, W/cm²) or fluence (J/cm²). Furthermore, the nomenclature should be harmonized, recognizing that power density is equivalent to irradiance.

Referencing Lee et al. (2019), it is noted that the reported fluence range is 0.3-3 J/cm². However, the table lacks corresponding parameter name.

Where both exposure duration (seconds) and fluence are provided, the author could consider calculating irradiance using the formula: Irradiance (W/cm²) = Fluence (J/cm²) / Exposure (seconds).

Validity of the findings

No Comment

Additional comments

1.Please consider revise the section titled 'Synthesis of Results' to 'Result Table Generation.' This change aims to clarify the process and mitigate potential misinterpretations.
2.The Discussion and Conclusion sections are very well written. These sections demonstrate a robust structure characterized by clear logical argumentation and effective compartmentalization of content.

---

## Round 0.2 · Minor Revisions

Please address the remaining reviewer comments.

·

Basic reporting

All my comments have been dealt with satisfactorily.
I recommend that the publication be accepted in its present form.

Experimental design

see first report

Validity of the findings

see first report

Additional comments

see first report

·

Basic reporting

no comment

Experimental design

no comment

Validity of the findings

no comment

Additional comments

The revision effectively reflected the review comments provided.
However, some parameter names in table 2 are still missing, and would be ideally completed to insure readability.
For example for Assis et al 2012, please revise the dosage column into
" wavelength: 808nm, power: 30mW, fluence: 180J/cm², irradiance 3.8W/cm², total energy: 1.4J "
or the alternative parameter names author deemed to be appropriate.
Please double check row" De souza et al 2011" and "Alves et al 2012 " dosage column to label "wavelength" within in the content.

---

## Round 0.3 · accepted · Accept

The revised manuscript has been examined by one referee and myself. Concerns raised in the first round of review have been satisfactorily addressed.

·

Basic reporting

-

Experimental design

-

Validity of the findings

-

Additional comments

-